# Faces in the crowd: Twitter as alternative to protest surveys

**Christopher Barrie**[1]\* , **Arun Frey**[2,3]

**1** School of Social & Political Science, University of Edinburgh, Scotland, United Kingdom, **2** Department of Sociology, University of Oxford, England, United Kingdom, **3** Leverhulme Centre for Demographic Science, Oxford, England, United Kingdom

These authors contributed equally to this work and are listed in alphabetical order.
\* christopher.barrie@ed.ac.uk

## Abstract

Who goes to protests? To answer this question, existing research has relied either on retrospective surveys of populations or in-protest surveys of participants. Both techniques are prohibitively costly and face logistical and methodological constraints. In this article, we investigate the possibility of surveying protests using Twitter. We propose two techniques for sampling protestors on the ground from digital traces and estimate the demographic and ideological composition of ten protestor crowds using multidimensional scaling and machine-learning techniques. We test the accuracy of our estimates by comparing to two in-protest surveys from the 2017 Women's March in Washington, D.C. Results show that our Twitter sampling techniques are superior to hashtag sampling alone. They also approximate the ideology and gender distributions derived from on-the-ground surveys, albeit with some bias, but fail to retrieve accurate age group estimates. We conclude that online samples are yet unable to provide reliable representative samples of offline protest.

## Introduction

Writing at the close of the revolutionary Nineteenth Century, Gustave Le Bon [1, 15] saw a French society undergoing transition. And among "the most striking characteristics of our epoch of transition," he wrote, was the entry of the crowd into politics. But how to understand crowds? For his part, Le Bon [1, 23] claimed to find some element of "mental unity" among crowd members. Unwilling to cast the crowd as a singular entity, George Rudé [2], would later set out to identify the "faces in the crowd," and to give names and significance to individual crowd members. Giving life to individual crowd members, however, was a serious undertaking. This was because "participants. . . rarely leave records of their own," meaning the historian had to play archaeologist of revolt, piecing together whatever documentary evidence remained [2, 12]. Often, even where they were available, such records would not survive the impassioned context of their creation: the French National Archives, founded in 1790 to prevent the revolutionary destruction of public records, were later set aflame in the last weeks of the 1871 Paris Commune.

**Data Availability Statement:** All raw data files underlying the analysis are available at Harvard Dataverse (DOIs: 10.7910/DVN/5ZVMOR and 10.5683/SP/ZEL1Q6). Anonymized scripts and data used to generate the final analysis datasets, as well as curated, anonymized datasets to reproduce the figures, are provided on OSF as a public project

(https://osf.io/ybtsd), and is also accessible via the project's DOI (10.17605/OSF.IO/YBTSD). We are unable to provide the raw tweets used as our data source due to the 'Content Redistribution' conditions set out by Twitter here (https://developer.twitter.com/en/developer-terms/) agreement-and-policy, which permit only the resharing of Tweet IDs. The decision not to provide the already-filtered tweet IDs of protestors identified at marches was made on the basis of respecting user privacy, and was agreed in advance with the University of Oxford Central University Research Ethics Committee (CUREC—IRB Equivalent). The Reference Number of this Ethics Decision is: SOC_R2_001_C1A_20_16. The point of contact for this decision, and the individual to whom data requests may be sent is: Agnieszka Swiejkowska, DREC Secretary, Department of Sociology, University of Oxford, 42-43 Park End Street, Oxford, OX1 1JD. Email: research@sociology.ox.ac.uk; Tel.: +44 1865 286177.

**Funding:** Arun Frey was supported by the UK Economic and Social Research Council (ESRC) and the German Academic Scholarship Foundation. The funders had no role in study design, data collection and analysis, decision to publish, or preparation of the manuscript.

**Competing interests:** The authors have declared that no competing interests exist.

Subsequent generations of scholars have relied upon general population surveys to make inferences about participants in protesting crowds. Three problems often accompany this approach, which we summarize as: 1) question generality, 2) small positive n, and 3) preference falsification. General population surveys sometimes ask questions about protest participation but questions are often too vague to learn about the correlates of participants in particular protests. Surveys fielded after major protest events that do target particular protests commonly capture only a tiny fraction of actual participants. They are also limited by two types of response bias: to take one example, when a mass mobilization event is successful, respondents are likely to claim participation as the socially desirable response, even if they did not participate [3]; conversely, when such mobilization fails, or participants are mobilizing counter to the initial protests (i.e., are counter-revolutionaries in the context of revolution), respondents may choose not to disclose participation for fear of repression or retribution [4, 5].

The other option is to survey protesters in the field with in-protest surveys. This technique faces three further problems, summarized as: 1) sample selectivity, 2) non-response bias, and 3) logistical constraints. In-protest surveys select on the dependent variable, making it difficult to arrive at larger population-level inferences, and are undermined by considerable problems of non-response [6]. What is more, the protest cascades that precipitate uprisings are not flagged in advance and often come as a surprise to participants and onlookers alike [7]. As a consequence, researchers rarely have the time to organize survey questionnaires, gain clearance from institutional review boards, and hire interviewers before streets once again empty. Finally, both general population and in-protest surveys pose financial costs that are prohibitive for most researchers without sources of external funding.

Given these constraints, and the increasing visibility of protest and dissent online, scholars have innovated by using social media as sources of information. Most often, social media researchers will sample data by the identifying hashtag associated with a protest or campaign [8, 9]. We do not know, however, if users who share information about a protest online have the same ideological outlook or basic attributes as offline protestors.

In what follows, we set out two techniques for the identification of protestors on the ground sampled from their online traces. We implement this technique on a sample of individuals tweeting about the Women's March—a series of protest events held across multiple cities in the USA in the first month of 2017 to advance women's rights and protest the presidency of Donald J. Trump [10]. We first identify protesters on Twitter by locating individual Twitter users on march routes across ten US cities on the day of the protest. Using multidimensional scaling and machine-learning methods, we then estimate the ideological preferences of Twitter protestor-users, as well as their basic demographic characteristics. For the largest of these marches—in Washington, D.C.—we benchmark our demographic and ideology estimates against those from two in-protest surveys, as well as against estimates from a random sample of #WomensMarch hashtag users on the day of the protest. Finally, given the difficulties of obtaining a sufficiently large sample of geolocated users, we test the accuracy of a second technique for obtaining a protestor sample by manually coding photos shared by Twitter users in Washington D.C. on the day of the protests.

Our contribution is twofold. First, by elaborating techniques reliably to identify protestors on the ground from social media, we significantly improve on existing approaches that monitor only movement-specific hashtags. We show that by using this method we are able, with greater accuracy, to capture the ideological and demographic attributes of protestor crowds. Second, we evaluate these improved identification techniques by comparing them to benchmark data from protestors surveyed at protest marches. Here, we show that despite the improvements of our proposed technique, protestors who share information online still differ in systematic ways from the average protestor on the ground. Future research should build on

our proposals for identifying protestors from their online traces, which represent an obvious advance on sampling by hashtag alone. In turn, the viability of surveying protestors from digital traces alone will depend on future levels of connectivity and further advances in the automated inference of online user attributes. Taken together, our results at once provide avenues for further research and reason to be cautious when inferring movement information on the basis of digital traces alone.

## Surveying protest

Social movements and collective action constitute core fields within sociological research. And to pursue research in the field, scholars have made extensive use of both in-protest and retrospective surveys to understand the correlates of participation.

A first approach to gauging the correlates of participation involves using population surveys to capture both protestors and non-protestors in the sampling frame. Typically, such surveys are intended to be nationally representative. An early example is the work of Barnes and Kaase [11] who used population surveys to study attitudes toward protest across five Western democracies. Questions on protest participation have more recently been included in major cross-national surveys like the World Values Survey (WVS). Unfortunately, these questions are generally unspecific and therefore cannot accurately identify which type of protest the individual took part in or when it took place [12].

When a particular protest event *is* targeted within the survey design, researchers are often faced with the problem of a small positive n. By way of example, Wave II of the Arab Barometer surveys included questions on participation in the 2010–11 Arab Spring protests in Egypt and Tunisia—two large-scale mass-mobilization episodes. Despite the size of the Egyptian Revolution, only 8% of respondents (n = 97) reported participating [13]. Other examples do have a relatively large positive n [3, 5]. But [5] relied on a regular survey being fielded at the time of protest outbreak—the kind of chance coincidence on which researchers cannot rely. The "true" number of participants will often be smaller than the survey estimates: when mass mobilization events such as these are successful, asking retrospective questions about participation is subject to potential bias due to the "hero effect," whereby individuals claim participation despite the reality of their non-involvement [3]. Beissinger [5], for example, reports participation of 18.6% in the 2004 Orange Revolution in Ukraine; which would amount to 7.4 million people. This estimate would make the event one of the largest mass mobilizations in world history. Further, this bias runs in both directions. In the same study of Ukrainian protestors, Beissinger [5, 580] notes that "the number of counter-revolutionaries was likely twice as large as the [survey] indicated," since those protesting against the mood of the crowd are less likely to disclose their true preferences.

The other survey tool available to researchers is the in-protest survey. To date, the most ambitious project to use these methods has been "Caught in the Act of Protest: Contextualizing Contestation" (CCC) [14], an effort by researchers across Europe to understand the sociological underpinnings of protest through in-protest surveys at some ninety-two protest events across seven European countries [15, 16]. For the deployment of these instruments, researchers have also elaborated sophisticated random walk sampling frameworks to ensure the representativeness of the protestor sample [14, 17].

There are nonetheless several problems inherent to in-protest surveys. Most obviously, this method samples on the dependent variable, excluding non-protestors by design. What is more, conducting in-protest surveys poses another set of challenges. The collection of protest data can be (literally) noisy: in nearly half of the protest surveys they carried out, interviewers in the CCC Project reported having difficulty hearing their interviewees; in one fifth of cases,

interviewers reported difficulty given the chaotic nature of the demonstration, leading to increased non-response [6]. Delayed refusal caused by individuals not returning postal questionnaires was even more pronounced, leading these authors to conclude that "noncooperation is a serious problem in protest surveying" [6, 93]. Perhaps the biggest threat to this design, however, is the unpredictable nature of protest. Large-scale protest has a habit of breaking out all of a sudden [7]. This unpredictability necessarily confounds efforts to field survey teams at unexpected protest—for all protests covered in the CCC Project, protest organizers and police were contacted at least two weeks in advance of any action [14].

Against this backdrop, and the increasing visibility of protest on social media platforms, researchers have more recently started using digital trace data for the study of protest. The most common platform for this research, given both its accessibility and popularity for campaigning, is the micro-blogging service Twitter. Researchers in this area have used Twitter data to study the dynamics of protest movement mobilization [8, 9], recruitment [18], polarization [19], and change [20]. Two problems attend this research. First, using samples derived from online platforms can provide insights into *online* mobilization dynamics but the generalizability of these insights to the *offline* world remains conjectural. As Steinert-Threlkeld [8, 400] writes in his analysis of mobilization dynamics during the 2011 Arab Spring: "[the] article *assumes* that behavior on online networks parallels that of offline interpersonal ones" [emphasis added]. Similarly, given that both González-Bailón et al. [18] and Barberá et al. [9] rely on online samples alone, they are naturally able to suggest only that their findings might inform theoretical models of (offline) collective action. Second, different sampling techniques may yield different results. Most often, to arrive at their sample, practitioners will filter on a set of hashtags related to the given protest campaign. This is the case for all of [8, 18–20]. But as some the same practitioners have noted, different filtering techniques can generate very different samples when studying online protest communication [21]. Rafail [22] demonstrates in the case of the Occupy Wall Street (OWS) campaign, for example, that sampling on hashtag alone misrepresents the online network structure of the OWS movement, and underrepresents online mobilization activity. Of course, in the below, our starting point is also a "hashtag sample" but we go on to outline two different approaches for filtering these data to recover a sample of (offline) protestors on the ground. In summary: existing research has taken samples from online sources to make important insights about the dynamics of collective action. However, the question of whether samples sourced online correspond to the characteristics of offline samples has yet to be examined.

## Data and method

To fill this gap, we conduct two principal tests. The first compares our proposed techniques for capturing the digital traces of actual offline protestors to samples of users filtered by hashtag use alone; the second compares our Twitter-based sample of protestors to estimates from two in-protest surveys. In this, we are able to determine: 1) whether our proposed technique represents an improvement on more crude estimates from hashtag samples alone; and 2) the accuracy of our Twitter-based estimates when compared to the data from in-protest surveys.

To to build our dataset of protestors, we use two datasets of more than 8.6m tweets related to the 2017 Women's March. The first is taken from Littman and Park [23], which records tweets across several hashtags related to the Women's March; the second is taken from Ruest [24] and records tweets containing the hashtag #Womensmarch. The Littman and Park data was collected over the period December 19, 2016 to January 23, 2017 and the Ruest data from January 21, 2017 to January 28, 2017. The first sample we draw from these data is a random sample 5000 users who used one of the identifying hashtags on January 21, 2017. Included in

this sample were all users for whom we could recover ideology and demographic estimates. We call this our "Random" Twitter sample and use this as a benchmark against which to compare estimates derived from our proposed techniques for capturing actual protestors on the ground.

## Obtaining a sample of protestors

Identifying protest participants from the online behaviour of Twitter users alone is challenging: Protests often spark online commentary from participants, supporters, news reporters, and opponents alike. Those using the hashtag of a given protest may therefore be any of: 1) actual participants on the ground; 2) online supporters only; 3) online opponents only; 4) online commentators only.

To identify users who were posting on Twitter from within the march, we begin by filtering the tweet dataset to tweets sent on the day of the event (January 21, 2017). Our analysis began two years after the Women's March. We then filtered these data again to only those tweets that include location information in order to obtain digital traces of actual participants on the ground. Since only a small fraction of all Twitter users enable the geolocation of their tweets, this step considerably reduces our sample size from 3.8m to 17,120 tweets. To further restrict this data to actual protestors, our technique locates individual users to within a buffer of the protest march route on the day of the protest. To do this, we first sourced maps of the protest routes for ten of the largest protests during the Women's March online. A full list of the maps and their (archived) sources are S1 Table in S1 Appendix. Using the open-source geographic information systems software QGIS, these maps were georeferenced by locating landmarks and assigning relevant coordinates against reference coordinates from Open Street Map vector layers. Using this technique, we were able to obtain samples of protestors across all ten US cities. Inclusion in these samples relied on the user tweeting about the Women's March from within a 1km buffer of the march route on the day of the protest. Of all 17,120 tweets for which location data was available, we identified 2,569 unique users whose tweet(s) located them at one of the protest marches. S1 Fig in S1 Appendix provides a visualization of the end result of this process. We refer to this sample as our "Geolocated" Twitter sample.

Although the original Tweet ID datasets by Ruest [24] and Littman and Park [23] contained ~14.4 and ~7.2m tweets respectively, only around half could be recovered for each source likely due to either account deletion, tweet deletion, or user removal by Twitter. The latter is the least concerning for our purposes as removed accounts will be mostly bots. While we cannot be sure of the magnitude of bias introduced by the omission of users, we see no obvious reason for account or tweet deletion to introduce bias along demographic or ideological dimensions. It is possible that our Geolocated sample would have included ideological opponents to the movement in the vicinity of the protest who subsequently deleted tweets, either because they did not want to be associated with a minority movement or otherwise. Our Photo-coded sample screens for opponents and so would have removed these accounts, had they remained in the sample. Where such bias would have affected findings is in the Random (hashtag) sample, for which inclusion is based on hashtag use alone. Here, subsequent tweet deletions by more conservative users may have skewed the ideology distribution leftwards. While we cannot determine the size of this possible bias, it does provide further support for our argument the hashtag sampling alone is unlikely to recover a close approximation of offline protestor ideology and demographics.

Here, is also worth noting that by using geolocation as our sole inclusion criterion, we do not exclude potential commentators who are reporting from within the protest (i.e., journalists as opposed to protestors on the ground). In the S1 Appendix we discuss the size of any

potential bias caused by their inclusion. We first calculate the percentage of users in our geolo-cated samples who are "verified"—an indication that a user may be a journalist or news organi-zation in protest contexts—and then manually label a random subsample of our Washington D.C. geolocated tweets as "commentators" or "opponents." The percentage of users who are verified ranges from 0–7% across our ten cities. The percentage of tweets by commentators (rather than protestors) is ∼4% in our random Washington D.C. subsample; the percentage of tweets by opponents is.2%. Whether or not such individuals, who are "caught up" in a protest, satisfy inclusion criteria will depend on the research question at hand. In any case, exclusion of these accounts, on the basis of their verification status or (in the case of the Washington D.C. protest) manual codings, does not substantively alter our findings. As we detail below, we also evaluate a second, photo-coding, procedure for identifying protestors on the ground (where we screen for and exclude opponents and commentators) and are able to compare the findings from this approach to our results from the geolocation procedure for the Washington D.C. Women's March.

## Obtaining ideology estimates of protesters

For both our Random and Geolocated samples we then estimate for each user their position on an ideology scale using a novel method originally developed by Barberá [25], which com-putes ideology estimates of Twitter users by examining which political actors they follow (in Twitter parlance, their "friends"). This technique is broadly analogous to other multidimen-sional scaling techniques used to estimate the ideological leanings of individual legislators from roll call data [26]. However, in the place of voting, Barberá demonstrates that practition-ers can leverage information on the friends of individual users to estimate their ideological position on a latent underlying dimension.

At its core, this estimation relies on the assumption that a user, given a set of otherwise sim-ilar political Twitter accounts with varying ideological beliefs, will prefer to follow those accounts that closely match her own ideological position. This is because the decision to follow a political account is costly: following a Twitter user entails the opportunity cost of not being exposed to alternative sources of information, and may induce cognitive dissonance if that information is at odds with one's own ideological outlook [25].

Several multidimensional scaling techniques, including ideal point estimation and corre-spondence analysis, are suitable for estimating the ideology scores of individual users [27]. In this article, we use a correspondence analysis procedure, since it gives effectively the same results as the Bayesian ideal point technique outlined in [25] while being computationally more efficient [27].

To estimate the ideology scores of our Random and Geolocated users, we begin by down-loading the friends of each user using the Twitter REST API with the `rtweet` R package [28]. We then follow the procedure set out by Barberá et al. [27], using the R package "`tweet-scores`". This package includes a pre-specified list of US "elites" from politics and news media spanning a liberal-conservative dimension. We then estimate individual user ideology scores by first arraying a sparse adjacency matrix of individual protestor user (rows) and elite friends (columns) as in Fig 1.

It is then possible to project each individual user matrix $u$ back onto the latent ideological space already estimated by first taking the vector of the standardized residuals $u' = \frac{u}{\sum_i u_i}$ for each supplementary user then calculating the location of the new user on the latent ideological space $g = u'^T c$, where $c$ represents the vector of column coordinates for individual political elites. The "`tweetscores`" package is able efficiently to add users (or rows) to a correspon-dence analysis procedure without re-estimating the entire correspondence analysis. It does so

$$\begin{pmatrix} - & \text{@ABC}_1 & \text{@BarackObama}_2 & \text{@CynthiaLummis}_3 & \dots & \text{@zeitgeist2o12}_{1186} \\ \text{@WomensMarcher}_1 & 0 & 1 & 0 & & 1 \end{pmatrix}$$

$$\begin{pmatrix} - & \text{@ABC}_1 & \text{@BarackObama}_2 & \text{@CynthiaLummis}_3 & \dots & \text{@zeitgeist2o12}_{1186} \\ \text{@WomensMarcher}_2 & 1 & 0 & 0 & & 1 \end{pmatrix}$$

**Fig 1. Example adjacency matrices.**

by projecting the row coordinates of the new user onto the already-estimated latent ideological space by taking the row coordinates of the new user and looking up the corresponding column coordinates from a pre-estimated set of representative values. When the row coordinate does not have an exact match in this pre-estimated list of corresponding column coordinates, the function takes the closest corresponding column coordinate value and adds a value from a random normal distribution with mean 0 and standard deviation.05. This is why the estimated ideology score of each user will randomly vary by a small amount on each estimation. The estimation of a user's ideology score relies on her following network. Thus, if a user follows no elite accounts, their ideology score cannot be computed. For the Geolocated sample, this is the case for 111 observations, or 4.3% of the sample. We describe the reasons for different types of missingness in more detail in the S1 Appendix.

The estimation procedure also accounts for "user- and elite-random effects" by including parameters for the political interest of user $i$ (number of elites they follow) and the popularity elite $j$ (number of followers of elite). The former acts as a proxy for the political interest of the user (i.e., a user may follow many accounts because they are simply interested in politics) and the latter accounts for the fact that a user may follow popular Twitter accounts (e.g. Barack Obama) simply due to their high profile and general relevance rather than as a function of ideological proximity (see supplementary material [25] and [27]). We provide descriptive statistics on the number of elite accounts followed by users across the samples in the S1 Appendix.

## Obtaining demographic estimates of protestors

We next supplement our ideology estimates by inferring basic demographic information from the Twitter profiles of individual users [29]. Wang et al. [29] propose a deep learning system that assigns each Twitter profile a probability of being male or female and belonging to a specific age group ($\leq$18, 19–29, 30–39, 40+). To infer users' sex and age group, Wang et al. [29] relies on four sources of information from Twitter: the username, screen name, biography, and profile image of each user. Each of these sources of information is evaluated using a separately trained text- or image-based neural model, before being combined for classification into a shared pipeline. Combined text and image information for each user is then classified with using the "m3inference" library in Python. This estimation technique is preferable as it does not rely on large quantities of text produced by any individual user in order to generate demographic estimates, thus lowering computational costs. Despite its sparse input, the M3 model significantly outperforms state-of-the art techniques for inferring age and gender from image and text data. This includes "Face+++" [30], "Microsoft Face API" [31], "genderperformr" [32], "demographer" [33], and [34]. By not relying on text output, it is also scaleable to multiple languages other than English. We use this information to estimate the demographic composition of our sample. We were unable to recover demographic information for 148

users, or 5.8% of the sample. After removing the missing values for both ideology and demographic estimates, the Geolocated sample includes 2,319 unique users.

## Alternative sampling procedure

Only a very small subset of users provide precise geolocation coordinates. This is one reason that research to date has opted to use alternative location information to estimate protestor crowd size [35]. Recognising this constraint, we elaborated a second sampling procedure to capture protestors on the ground from their online traces. This second approach makes use of information contained in photographs shared by Twitter users. Given that the march in Washington D.C. saw the highest participation and we have in-protest survey evidence against which to compare our estimates, we only carry out this technique for Washington D.C. tweeters. To obtain a sample of protestors we first filtered our tweet dataset to users who posted original photographs and whose location ("Place") mentioned the city of Washington D.C, leaving 2,750 tweets. Twitter aggregates location to a Twitter "Place." Twitter Places can refer to a specific place (like a stadium or monument) or an aggregate geographical location such as a city. For more information on Twitter Places, see https://developer.twitter.com/en/docs/tutorials/filtering-tweets-by-location.

We code a user as having participated in the protest if: a) the photo was taken from within the protest crowd during the Women's March in Washington D.C.; and b) if the image and accompanying text indicated protest attendance. We exclude tweets indicating news reporting rather than actual participation, as well as photos that could be stock images. We include in the S1 Appendix of this article the full criteria that we used during the coding process. Each author independently coded half of the photographs dataset (∼1300 tweets containing photographs) and jointly coded a subset of 200 photograph tweets. A comparison of our respective codings generated an inter-coder reliability Cohen's Kappa score of 0.8, indicating substantial agreement. Of the 2,750 tweets that included original imagery, 1,125 were coded as having been taken by protest participants. With this photo sample, we then repeated the same steps outlined above to generate ideology scores and estimates of crowd demographics. We refer to this sample as our "Photo-coded" sample. In total, we were unable to recover ideology estimates 201 users and demographic estimates for 49 users, resulting in a final sample size of 922. We describe the reasons for different types of missingness in more detail in the S1 Appendix.

We summarize the entire workflow used to arrive at these estimations S2 Fig in S1 Appendix. The process detailed above results in three samples of Twitter users for whom we are able to recover ideological and demographic estimates. The first, Random sample of #WomensMarch hashtag users includes any user who posted with a relevant hashtag on the day of the protest; the second Geolocated sample includes any user identified on one of the protest routes across ten US cities; the third Photo-coded sample includes only users identified to the protest route in Washington, D.C.

## Ethics

Before embarking on this research, we took account of a large number of ethical considerations. We summarize below what we determined on the basis of these considerations, and detail in full the ethical framework according to which we approached this research in the S1 Appendix. First, we gained authorization for this design from our institution's Central University Research Ethics Committee (Institutional Review Board equivalent). We describe details of this ethics decision in the S1 Appendix. We did not obtain informed consent from "participants" in this research as this was not deemed necessary. Consent is assumed as data is publicly available. Nonetheless, with a view to preserving contextual integrity [36] and user anonymity

given the potential sensitivity of these data, we determined to: 1) elaborate an anonymization procedure prior to, and during, data ingestion to reduce any exposure to identifying information; 2) store all potentially identifying information locally on encrypted folders; 3) not to release tweet IDs of geolocated and photo-coded users in public replication folders.

## Results and validation

We first present the results from our geolocated users. Twitter-based estimates of crowd ideology distributions across ten US cities are depicted in Fig 2. We observe distributions centred to the left of ideological centre (depicted by a dashed grey line at 0). The distributions are very similar between cities, indicating a substantial degree of between-protest ideological homogeneity.

Our estimates of crowd demographics are displayed in Figs 3 and 4. Across most of our ten US cities, crowds are overwhelmingly female and tend to come, in the majority, from younger age groups. The exceptions are the cities of Portland and San Francisco, where the 30+ groups are in preponderance and there is almost gender parity. Both of these samples suffer from a very small n, however, and should therefore be treated with appropriate caution.

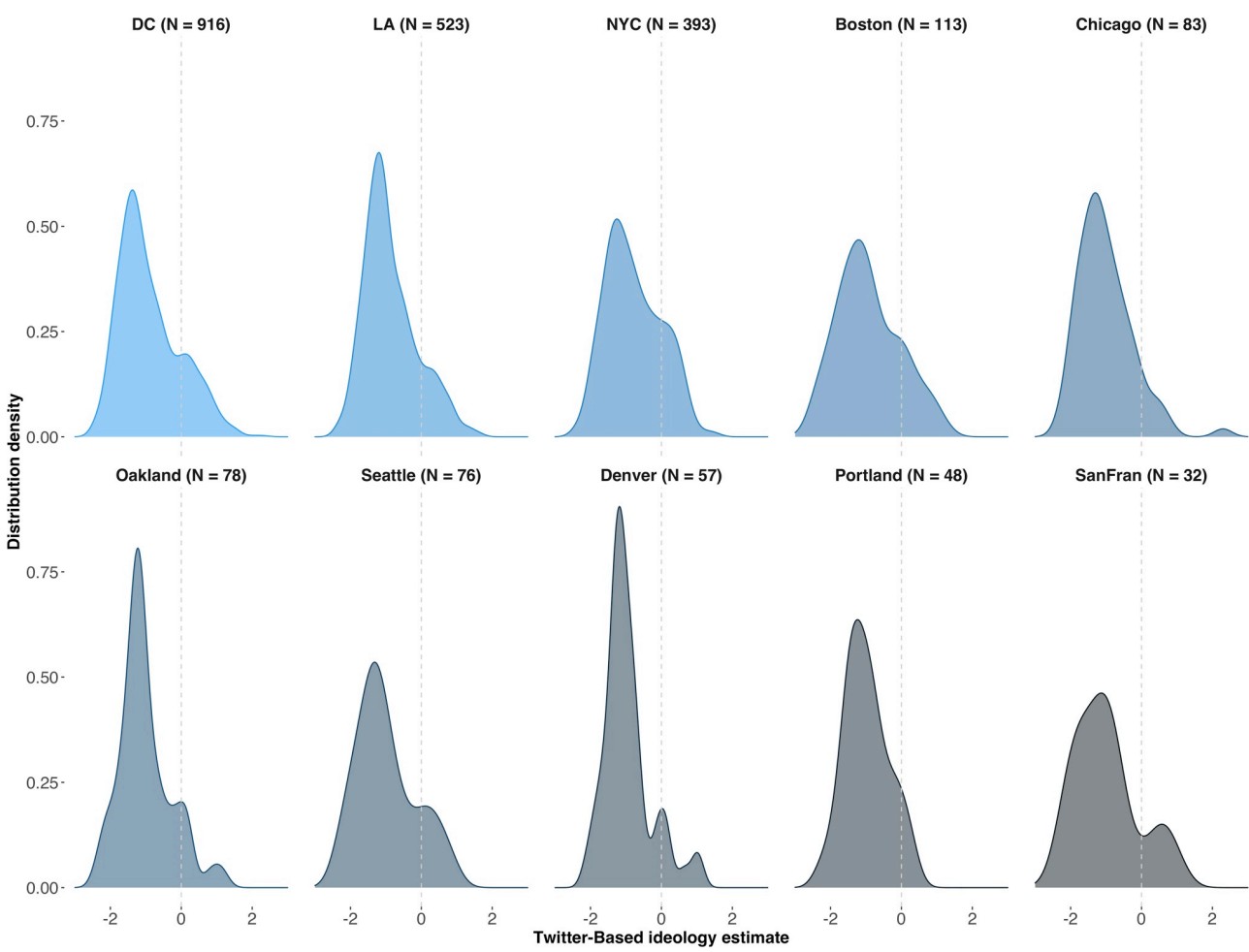

**Fig 2. Distributions of aggregate crowd ideologies across ten cities in the 2017 US Women's Marches from geolocated users.**

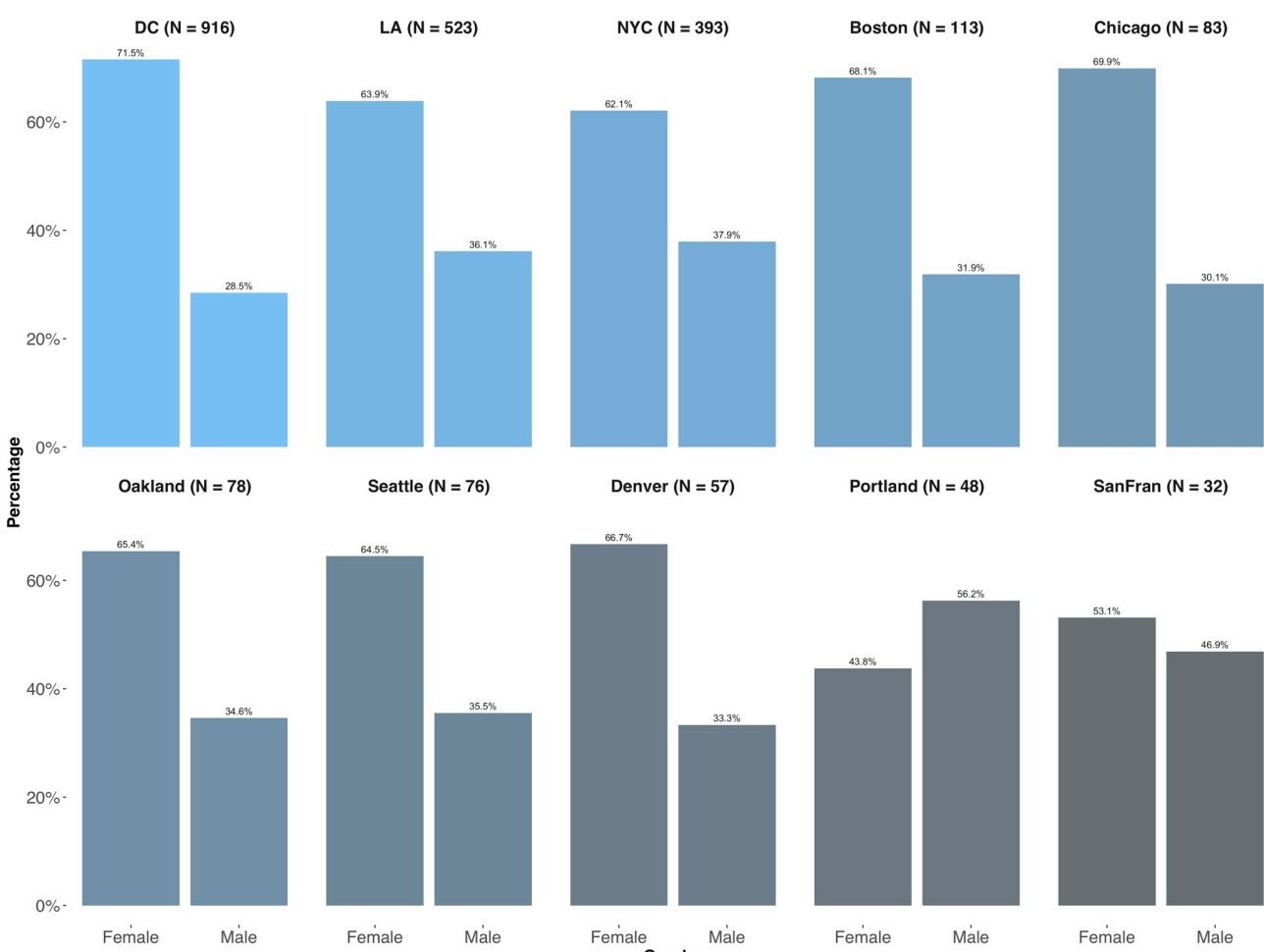

**Fig 3. Gender distributions in the Twitter-based samples across ten cities in the 2017 US Women's Marches from geolocated users.**

To scrutinise the validity of our results, we require a benchmark against which to compare them. These data are available for the Washington, D.C. Women's March where two in-protest surveys were conducted by Fisher et al. [37] and Heaney [38]. We focus first on the ideology estimates and second on demographic estimates. For the first, the in-protest survey asks participants to place themselves on an ordinal ideological scale, from "Very Left" (1) to "Very Right" (7) in Fisher et al. [37] and from "To the "left" of strong liberal" (1) to "To the "right" of strong conservative" (9) in Heaney [38]. While in-protest surveys are subject to their own biases, they nonetheless represent the gold standard for obtaining systematic data on protest participation. For this reason, we use these surveys as a benchmark for our own Twitter-based estimates. As we go on to describe below, these independent surveys also produced estimates for ideology and demographic distributions that closely correspond to each other. The refusal rates for both surveys were relatively low (7.5% for Fisher et al. [37] and 20% for Heaney [38]), and they both employed similar crowd sampling strategies. As such, we claim that these surveys constitute a valid and high quality point of comparison.

In addition to our protestor-users geolocated to Washington D.C., we now incorporate our two other Twitter samples for these comparisons. The first is our photo-coded sample of protestors at the march in D.C.; the second is our random sample of users filtered by hashtag

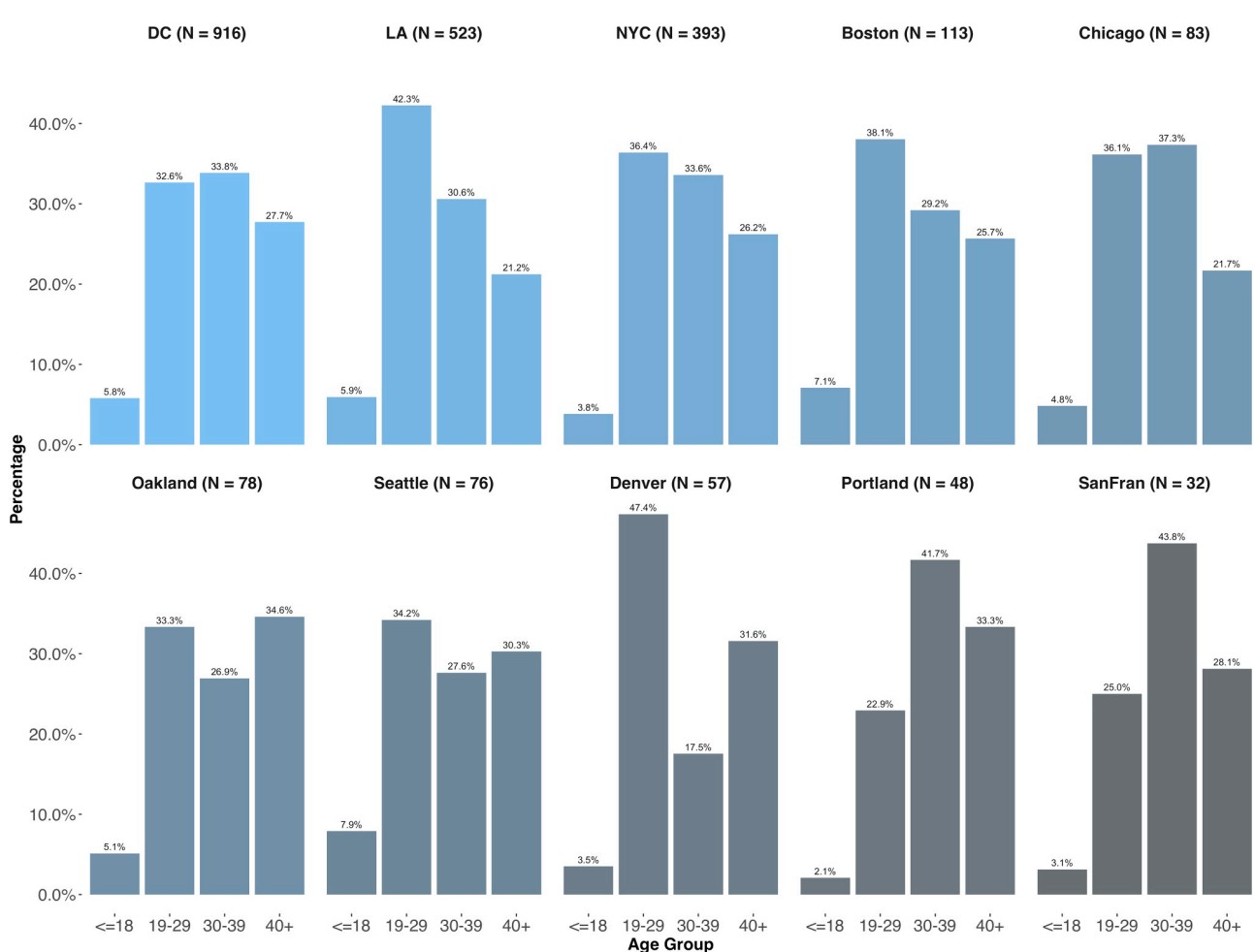

**Fig 4. Age distributions in the Twitter-based samples across ten cities in the 2017 US Women's Marches from geolocated users.**

alone. Note that the users in this second sample could be tweeting from any location and may or may not have attended the D.C. protest—inclusion was based solely on their having tweeted with the #WomensMarch hashtag on the day of the protests. We then compare the Twitter-based estimates of ideology distributions to survey results in Fisher et al. [37] and Heaney [38]. We visualize the distributions of ideology scores for the in-protest survey and Twitter samples in the upper panel of Fig 5. We only use observations with complete records for the purposes of comparison. The number of observations for each sample therefore represents observations for which we have complete records for age, gender, and ideology.

The Twitter-based ideology estimates are already standardized to follow a normal distribution with mean 0 and standard deviation 1; that is, a user with score -1 is to be understood as one standard deviation to left of the "average" user [27]. For the purposes of comparison, we centre the ideology scales of the in-protest surveys such that a score of 0 represents the middle category of each respective ordinal scale before standardizing by dividing by one standard deviation. The middle categories for each of the in-protest surveys are: (5)"moderate" in Heaney [38] and (4) "Moderate, middle of the road" in Fisher et al. [37]. We see that individuals surveyed in-protest are relatively more left-wing than our Twitter-based geolocated and photo-coded samples. Our Twitter-based samples of identified protestors nonetheless do have

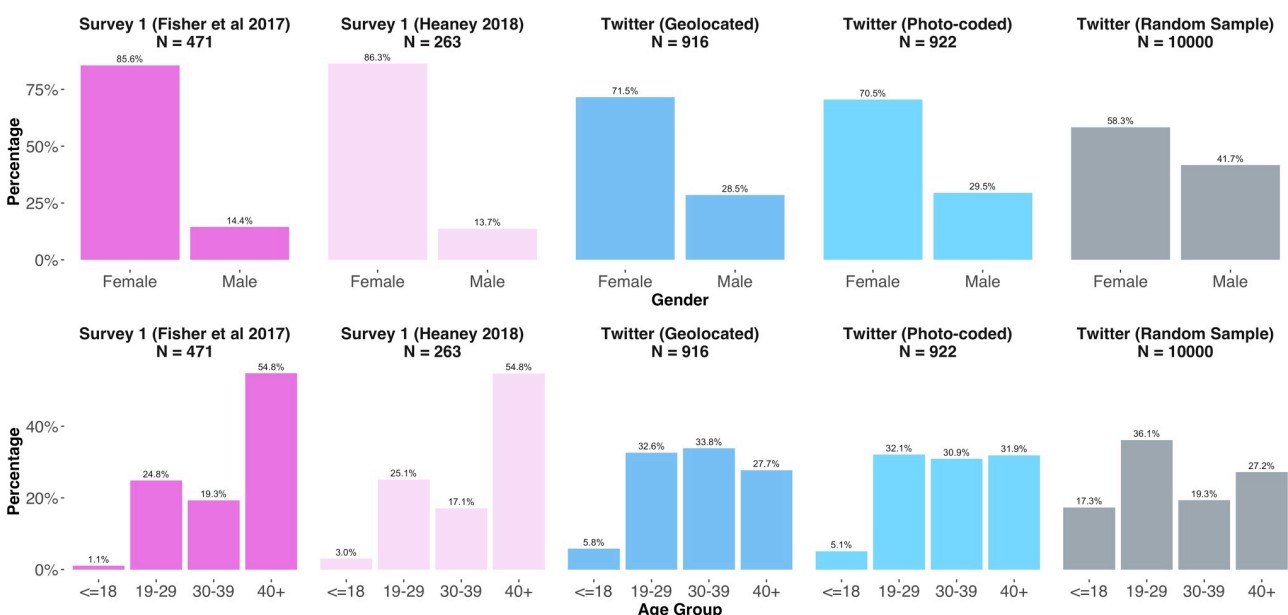

**Fig 5.** *Upper panel:* Ideology score distributions in survey- and Twitter-based samples for Washington, D.C.; *Lower panel:* Comparison of ideology distributions in protestor Twitter sample and random sample of all accounts using the hashtag `#WomensMarch`.

ideology distributions that are similarly right-skewed, peak to the left of zero, and have negative modal values.

It is important to note that, despite the different sampling strategies, both Twitter samples of geolocated and photo-coded protestors provide highly similar estimates of crowd ideology. To assess whether this similarity is merely a feature of the underlying data, we compare our Twitter-based ideology estimates with the ideology estimates obtained from our random sample of hashtag users. In the lower panel of Fig 5 we overlay the ideology distributions for our geolocated and photo-coded users on the distribution for the random sample. We see that the estimates for those users we identify as protestors on march routes have greater density to to the left of zero than our estimates for the random sample. This is initial evidence that simply using hashtags to identify protestors is insufficient for capturing the ideology distributions of actual protestor crowds, and suggests that both geocoding and photo-coding methods identify similar users as protestors. Note that only 46 users are in both the geolocated and photo-coded Twitter samples. This means the similarity between both samples is not due to considerable overlap in users who geolocated themselves at the protest march, and users who uploaded a tweet containing a photo from within the protest.

Next, we compare the demographic estimates from our Twitter-based samples to those derived from the in-person surveys (Fig 6). We can see that in both the geolocated and photo-coded Twitter samples, similar to the in-person surveys, there is a preponderance of women making up the crowd. The geolocated and photo-coded Twitter-based samples are highly similar across both age and gender composition; compared to the in-protest surveys, however, they feature substantially more male participants, with male users making up 28.5% and 29.5% of the Twitter samples versus 14.1% and 16.2% respectively for the Fisher et al. [37] and Heaney [38] samples. Age differences between the online and in-protest samples are more pronounced. The modal age group in the geolocated and photo-coded Twitter samples is 19–29, for example, whereas for the in-protest sample it is the 40+ group. Still, across both age and

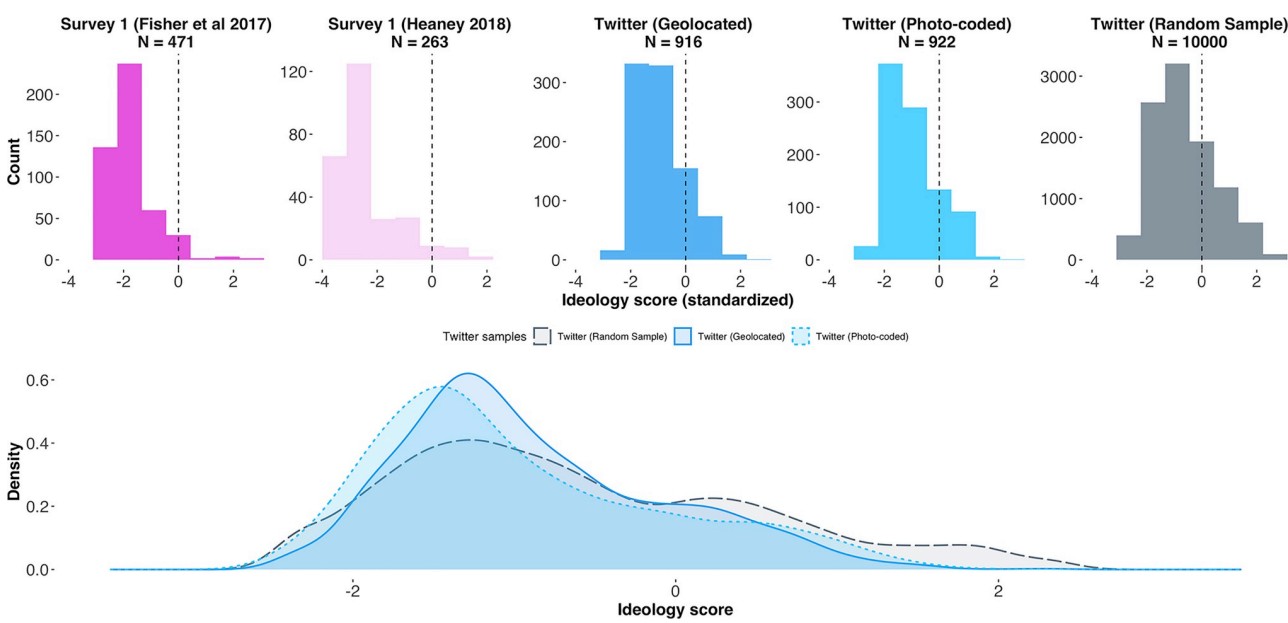

**Fig 6. Gender and age distributions in survey- and Twitter-based samples for Washington, D.C.**

gender, the geolocated and photo-coded samples of Twitter users look alike and improve on the gender estimates derived from the random Twitter sample.

## Discussion and conclusion

The use of digital trace data to make inferences about crowds has, to date, largely focused on the estimation of crowd size [35, 39]. This paper represents the first test of the viability of using online digital traces to estimate demographic and ideological characteristics of protestor crowds. For this, we rely on the availability of two in-protest surveys against which to compare the Twitter-based estimates. We find that Twitter can provide approximations of the ideological and gender composition of crowds but there remain considerable biases. Irrespective of sampling strategy, we are unable to recover accurate estimates of crowd age demographics.

What explains these differences? One explanation could be differences in the type of person likely to post about protest participation online. Geolocating to a particular event entails privacy costs, which digitally literate users may be less likely to accept. It is nonetheless worth noting that our geocoding and photo-coding techniques for identifying protestors do give very similar estimates despite the different sampling procedures used. This suggests that both methods do well to capture protestors on the ground who are also active online. Another explanation is difference in measurement context. It may be that in the politically charged environment of a protest, individuals are more likely to place themselves further to the extremes of an ideology scale than they otherwise would have. Alternatively, bias may result from the inferential procedure used to derive ideology scores from follow networks. Some individuals follow only a few relevant accounts, meaning their ideology scores can only be estimated with error. That said, removing accounts who only follow a few elite accounts does not substantively alter the distribution of ideology scores in our protestor crowds (see S1 Appendix).

In the case of age and gender, bias may result from measurement error in the automated procedure used to infer these demographic characteristics. Importantly, this measurement error may be systematic. For example, younger or more digitally literate users may be more

likely to use an avatar in place of a photograph of themselves, thereby limiting the accuracy of algorithmic age and gender prediction. Then again, the large majority of our photo-coded sample used profile photos that did not appear to hide their real identity.

While we do not discount the above sources of potential bias, we suggest that the majority of the difference between our survey-based and Twitter-based demographic estimates most likely comes not from our sampling strategy or from classification error but from biases in the type of individual who is active on Twitter. After all, Twitter is not a representative sample of the general population. In the United States, the average Twitter user is younger, more likely to be male, and wealthier than the US populace [40]. What is more, political discussions tend to feature men more than women and disproportionately include more educated users and users from urban or metropolitan areas [41]. And the differences between our Twitter-based and survey-based demographic estimates map closely onto these sources of bias.

This notwithstanding, the findings do point to the potential future use of digital trace data as a source of information on the composition of protestor crowds. As connectivity and online platform usage increase over time, it is possible that these sources will become more representative of general populations [42]. What is more, we know that, even if the average Twitter user has a different demographic profile compared to the general population, they are nonetheless very similar on various attitudinal measures [43]. This insight accords with our own findings above, which show that, as a source of information on aggregate ideological preferences, Twitter provides estimates that approximate those from surveys on the ground.

As for the viability of this method in other contexts, we are less optimistic. In many respects, the Women's March protests represent one of the most-likely cases for recovering representative samples of protest crowds from digital trace data. After all, these were very large protests in a democratic setting with high connectivity. In other contexts, low connectivity will likely mean insufficient sample sizes. Further, in non-democratic political contexts individuals may be less willing publicly to signal dissent online. A growing body of work is nonetheless making use of digital trace data, and Twitter in particular, for the study of movement campaigns outside of Western or liberal-democratic contexts [44–46]. Validating the offline representativeness of users sampled online will require benchmarking to in-protest surveys conducted in these contexts (e.g., [5, 47–49]).

Several limitations of the technique presented in this article do highlight possible avenues for extending and refining the approach. First, the technique we propose uses data from only one platform. For future implementations of the basic method, our technique is by no means limited to Twitter, however. Gathering information on the ideologies of users requires only that the researcher can access relevant information on the accounts followed by any given user. On Facebook, this is equivalent to an account "liking" the page of a particular prominent individual; Instagram and Sina Weibo have a following option very similar to Twitter; VKontakte provides information on the "Groups" and "Public pages" of which any given user is member; and on both TikTok and YouTube, the equivalent would be subscriptions. As for collecting information on the gender and age of a user, this can be achieved using a neural architecture that relies only on limited user-level information, all of which would be accessible across diverse platforms.

Second, our method relies only on information that has been made publicly available by the user (i.e., their tweets, who they follow, their photo, user name, screen name, and account description). Naturally, this limits the amount of information the researcher is able to glean from any one individual. One future direction for the sampling method we outline would involve sending online questionnaires to sampled users. In order to shed further light on the sources of difference between offline and online samples, in-protest surveys might also ask for the Twitter handle of protestors. Researchers could then link the survey and Twitter data to

determine the correlates of online presence and activity in the context of protest. These methods would likely encounter high refusal rates, however, and have associated privacy concerns [50, 51]. We are also inferring age and gender algorithmically in the approach we outline, which entails measurement error—particularly for age [52]. An alternative would involve manual annotation by individual researchers, users themselves, or crowd-sourced online workers (see e.g., [53]).

Still, while our sample of online protestors is not representative of crowds on the ground, it does allow for within-platform comparisons. Digital trace data is "always on" [54], enabling researchers to construct longitudinal panels after the initial sampling frame has been established [45]. Differentiating between users who do and do not participate in protests also allows researchers to make use of a ready-made comparison group against which to benchmark their findings. Using digital traces to identify protest participants can thus help us understand how protestors' activity on social media platforms differs from other users, and can shed light on whether participation in a protest changes online behaviour over time.

Overall, this article provides a first validation test for using Twitter to "survey" protestors from afar. By locating users to the march route on the day of the protest, we identify protest participants on Twitter and compare their ideological and demographic composition to estimates from two separate in-protest surveys. Our method considerably improves on a random sample of all users tweeting about the #WomensMarch, and can recover an approximation of the ideological and demographic profile of protest crowds. Still, important differences remain between online and offline protestors: in line with general discrepancies between Twitter and the US populace, online protestors tend to feature a higher share of young and male participants. By signalling the capabilities and limitations of Twitter data for protest research, our results provide an important reference point for researchers wishing to study offline mobilization with online digital trace data.

## Supporting information

**S1 Appendix.**
(PDF)

## Acknowledgments

We thank Pablo Barberá, Michael Biggs, Neil Ketchley, Joshua Tucker, and Megan Metzger for comments on versions of this paper, Dana Fisher and Michael T. Heaney for making available survey replication data, as well as audiences at the 2020 American Political Science Association Online Conference and Nuffield Online Sociology Seminar.

## Author Contributions

**Conceptualization:** Christopher Barrie, Arun Frey.

**Data curation:** Christopher Barrie, Arun Frey.

**Formal analysis:** Christopher Barrie, Arun Frey.

**Investigation:** Christopher Barrie.

**Methodology:** Christopher Barrie, Arun Frey.

**Validation:** Christopher Barrie, Arun Frey.

**Visualization:** Christopher Barrie, Arun Frey.

**Writing – original draft:** Christopher Barrie, Arun Frey.

**Writing – review & editing:** Christopher Barrie, Arun Frey.

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
