## [Decision Letter · Decision Letter 0]

20 May 2021

PONE-D-21-07964

Faces in the Crowd: Twitter as Alternative to Protest Surveys

PLOS ONE

Dear Dr. Frey,

Thank you for submitting your manuscript to PLOS ONE. After careful consideration, we feel that it has merit but does not fully meet PLOS ONE’s publication criteria as it currently stands. Therefore, we invite you to submit a revised version of the manuscript that addresses the points raised during the review process.

the paper needs a MAJOR REVISION. Please, follow the suggestions given by reviewers in order to improve the quality of the manuscript.

We look forward to receiving your revised manuscript.

Kind regards,

Barbara Guidi

Academic Editor

PLOS ONE

Journal Requirements:

3. We note that Figures in the Appendix of your submission contain map images which may be copyrighted. All PLOS content is published under the Creative Commons Attribution License (CC BY 4.0), which means that the manuscript, images, and Supporting Information files will be freely available online, and any third party is permitted to access, download, copy, distribute, and use these materials in any way, even commercially, with proper attribution. For these reasons, we cannot publish previously copyrighted maps or satellite images created using proprietary data, such as Google software (Google Maps, Street View, and Earth). For more information, see our copyright guidelines: http://journals.plos.org/plosone/s/licenses-and-copyright.

3.1.    You may seek permission from the original copyright holder of Figures in the Appendix to publish the content specifically under the CC BY 4.0 license. 

3.2.    If you are unable to obtain permission from the original copyright holder to publish these figures under the CC BY 4.0 license or if the copyright holder’s requirements are incompatible with the CC BY 4.0 license, please either i) remove the figure or ii) supply a replacement figure that complies with the CC BY 4.0 license. Please check copyright information on all replacement figures and update the figure caption with source information. If applicable, please specify in the figure caption text when a figure is similar but not identical to the original image and is therefore for illustrative purposes only.

Reviewers' comments:

Reviewer's Responses to Questions

**Comments to the Author**

1. Is the manuscript technically sound, and do the data support the conclusions?

Reviewer #1: Yes

Reviewer #2: Yes

2. Has the statistical analysis been performed appropriately and rigorously? 

Reviewer #1: I Don't Know

Reviewer #2: N/A

3. Have the authors made all data underlying the findings in their manuscript fully available?

Reviewer #1: Yes

Reviewer #2: Yes

4. Is the manuscript presented in an intelligible fashion and written in standard English?

Reviewer #1: Yes

Reviewer #2: Yes

5. Review Comments to the Author

Reviewer #1: The authors of manuscript PONE-D-21-07964 propose a novel method for analyzing political leanings and demographics of protest participants by using twitter posts. For this, the authors located people who posted in close proximity to a protest route and analyzed these tweets in terms of political leaning, age, and gender. Results are compared to on the ground surveys and a random sample of tweets using hashtags alone. While the study makes an important contribution to advancing the understanding of the composition of protest crowds, I have a number of concerns, which I will now list below:

1. While the historical example in the beginning is a nice introduction to the topic, it could be shortened at bit.

2. The related literature section focuses a lot on surveys on protesters done on the ground, and gives only little information on related studies that used twitter hashtags to study protesters. The section could be improved by discussing the latter in more detail, elaborating on potential difficulties with this design. In doing so, the reader could understand even better why the novel method is superior to past studies that only used twitter hashtag data.

3. You state that by looking at the hashtag alone, you could not differentiate between 1) actual participants on the ground; 2) online supporters only; 3) online opponents only; 4) online commentators only. Using geo-tagging is an efficient way for determining whether a user has actually posted in close proximity to the protest route. However, how can you be sure that the user is not an on the ground commentator or an on the ground opponent? Please explain whether you were somehow able to control for this.

4. The number of tweets in the final samples are not included in the main text. Adding that information would add to the understanding of your procedure. In general, it is not obvious from the text how exactly the original sample of more than 8.6m Tweets was reduced to the final numbers of n = 916 for the geolocated and n = 922 for the photo-coded samples. Please explain this in more detail.

5. In the discussion, you state that the analysis was limited to the information users made publicly available. Can you provide more details about what public information was used exactly, and what shares of users could not be included because of this?

6. The benchmark sample was drawn at random from users who used the identifying hashtags. The two datasets that were used for this collected tweets over longer time periods, and not only on the day of the protest. For the geo-located sample, however, I understand that only tweets that were posted on the day of the protest were used. Wouldn’t the two methods be more comparable if for the random sample, too, only tweets that were posted on the day of the protest would have been considered?

In summary, I believe this study is well written and makes an important contribution to the study of protest participants. I would like to thank the authors for their hard work and wish them all the best for their subsequent steps.

Reviewer #2: The paper seeks to fill a very specific hole in our methodological knowledge: the degree to which Twitter samples (and, perhaps by implication, similar digital trace data) approximates traditional survey methods when seeking to understand protests. I think it is intriguing, and broadly of substantial use to those who research online and offline social movements, and is deserving of publication. I have some suggestions of ways in which the manuscript might be improved.

The use of the term “revolutions” is a bit confusing in places, as it is used differently across fields. E.g., in line 37, I suspect the authors are seeking a point that is closer to questions of tipping points toward mass violence or direct action, rather than a full-scale overthrow of the existing order, which is implied by the term “revolution.”

Leaving aside the difficulty of geocoding tweets more broadly, the authors should note whether they sought to categorize those who were incidentally in the location of protests. Large-scale protests are often in crowded urban environments, and those tweeting may be affected by the protests while not willfully participating in them. Naturally, whether those who are so caught up in protests should be included in a sample is a question of the research design and the research question at hand. But the assumption that the samples within a km of key central positions of the protests seems like it would catch a lot, most of the routes running through the densest business districts in each city, by design.

Footnote 9. While it is reasonable to assume removal by Twitter was largely of bot accounts, it also seems likely that non-bot accounts were removed, and likely such removals weighed heavily toward the conservative side. By your own reasoning, we might likewise assume that tweet or account deletions were more likely to be by conservatives wishing to disassociate themselves with a “losing” political movement at some point after the 2019 elections.

It’s a bit easy to lose track of the process here. The figure in the appendix helps a bit here, but I should be able to easily discern this flow in the text as well. Am I correct in understanding that the hashtag sample was your starting point and the geolocated and photo samples were filtered from the hashtag (“random”) sample? How and when was the random sample obtained? Given the flat number, how was it sampled?

The report of missing ideological scores (line 215) feels out of place given the lack of representation in the text of the n of the sample size. The percentages in Table 1 make this clearer, but the n for each of the samples should be highlighted early on (and likely listed in Table 1 as well, as total n for each sample). Relatedly, am I to assume you assigned an ideological score with any number of follows of the ideologically inflected accounts more than zero? So that if someone, for example, follows Barack Obama, but no other listed account, they are coded (presumably as liberal)? I can follow this up in the cite, but it feels dangerous.

The use of in-protest survey data as a comparator is perfectly reasonable, but calling these “ground truth” (ln 276) is problematic for many of the reasons you have already noted earlier in the manuscript. Protest survey data come with their own significant biases. One might make some guesses as to systematic error here that run pretty close to things like demographic data. Even in a relatively safe setting, women may be less likely to answer unsolicited questions by someone approaching them in public, for example. But more broadly, these are multiple attempts to ascertain a ground-truth that each approach is attempting to approximate. Comparing the multiple attempts against one another is natural, but assigning the survey data as the gold standard would require you to more clearly indicate why you are assuming this to be the case.

I appreciate the ethical note, particularly in the extended version appendix B, which is thorough and well-reasoned.

I feel like this drops things off rapidly at the end. This may be a matter of the brevity of the report. However, the question of (for example) differences in age could be explained in multiple ways: most especially, differences in age distribution among twitter users (or SM users mid-protest, more specifically) or a classification model that introduces systematic errors in approximating age. The latter could be ascertained by surveying twitter users for their age, either online (though tracking twitter users to survey them can be tricky and invasive) or by surveying protestors in person to ascertain whether they are tweeting (and potentially linking samples directly). In any case, it would be very helpful for the discussion to open up possibilities of extending the work undertaken thus far.

Overall, the alignment between appendices, labelled numerically in the manuscript but then alphabetically in the separate appendix document, is confusing and should be revisited.

6. PLOS authors have the option to publish the peer review history of their article (what does this mean?). If published, this will include your full peer review and any attached files.

Reviewer #1: No

Reviewer #2: No

---

## [Author Response · Author response to Decision Letter 0]

29 Sep 2021

We sincerely thank the editor and reviewers for giving us the opportunity to revise our manuscript. We have responded to the specific editorial comments in our response letter. Here, we take the chance to note firstly that we have formatted the article according to PLOS One requirements, detailed the limitations to, and justification for, the replication data we are able to share, removed copyrighted image content, generated a track changes version of the manuscript highlighting where edits have been made, and provided details of an anonymized replication repository containing the scripts and datasets required to reproduce our findings.

Reviewers can access anonymized replication materials at the following Open Science Framework Project: https://osf.io/ybtsd/ or via the project's DOI identifier 10.17605/OSF.IO/YBTSD. 

Our responses to the specific reviewer comments follow.

REVIEWER 1:

"The authors of manuscript PONE-D-21-07964 propose a novel method for analyzing political leanings and demographics of protest participants by using twitter posts. For this, the authors located people who posted in close proximity to a protest route and analyzed these tweets in terms of political leaning, age, and gender. Results are compared to on the ground surveys and a random sample of tweets using hashtags alone. While the study makes an important contribution to advancing the understanding of the composition of protest crowds, I have a number of concerns, which I will now list below:

1. While the historical example in the beginning is a nice introduction to the topic, it could be shortened at bit.

We agree and have shortened this introductory section, trimming away some of the extraneous detail and limiting it to one paragraph that describes previous generations of historical scholarship. 

"2. The related literature section focuses a lot on surveys on protesters done on the ground, and gives only little information on related studies that used twitter hashtags to study protesters. The section could be improved by discussing the latter in more detail, elaborating on potential difficulties with this design. In doing so, the reader could understand even better why the novel method is superior to past studies that only used twitter hashtag data."

We thank the reviewer for this observation. We have now expanded this final paragraph of the literature review, detailing some of the problems encountered by hashtag sampling, which we split it into two related issues: 1) the disjuncture between the online and offline when using hashtag samples to study protest; and 2) the bias induced by using hashtags alone (without further filtering) to study protest movements.

"3. You state that by looking at the hashtag alone, you could not differentiate between 1) actual participants on the ground; 2) online supporters only; 3) online opponents only; 4) online commentators only. Using geo-tagging is an efficient way for determining whether a user has actually posted in close proximity to the protest route. However, how can you be sure that the user is not an on the ground commentator or an on the ground opponent? Please explain whether you were somehow able to control for this."

This is an astute point, which we neglected to answer in the original manuscript. We have now added an Appendix section (“Further sample characteristics”) where we detail the percentage of accounts in our geolocated sample that are classed as “verified” users. Verified users have a blue tick by their name and are most often the accounts of news media organizations, journalists, and sometimes other well-known figures. The percentage verified across all our cities is relatively small (~0-7%) and removing them from the analysis does not make any substantive difference to our findings. In addition, we took a random subsample of 500 tweets from the DC geolocated sample and manually coded these accounts, on the basis of the tweet content, account description, and verification status, for whether or not they were journalists or opponents. We find that ~4% of tweets are from journalists. Only one tweet (.2% of the subsample) came from an opponent, and even this coding decision was a marginal one. These findings are also described in the Appendix section “Further sample characteristics” and referred to in a footnote at the end of subsection “Obtaining a sample of protestors” in the main text. Finally, it is worth noting that the manually coded photo-coded sample does filter out commentators and opponents explicitly in our coding criteria. That we obtain comparable estimates to our geolocated sample should therefore aid confidence in our findings. 

"4. The number of tweets in the final samples are not included in the main text. Adding that information would add to the understanding of your procedure. In general, it is not obvious from the text how exactly the original sample of more than 8.6m Tweets was reduced to the final numbers of n = 916 for the geolocated and n = 922 for the photo-coded samples. Please explain this in more detail."

Thank you for alerting us that this is not clear. In the main text section entitled “Obtaining a sample of protestors,” as well as the three subsequent sections, we now describe more precisely how we filtered the original Twitter data to arrive at our final sample sizes, as well as the sources of any missingness. We have also now introduced an additional Appendix section entitled “Missingness,” which describes the source of any missingess in our data and where observations were dropped. Finally, we include an updated workflow figure in the Appendix, which more clearly explains how we came to the final sample sizes for each of our analyses. 

"5. In the discussion, you state that the analysis was limited to the information users made publicly available. Can you provide more details about what public information was used exactly, and what shares of users could not be included because of this?"

We now make clear that we are referring to here to a user’s tweets, who they follow, their photo, user name, screen name, and account description.

"6. The benchmark sample was drawn at random from users who used the identifying hashtags. The two datasets that were used for this collected tweets over longer time periods, and not only on the day of the protest. For the geo-located sample, however, I understand that only tweets that were posted on the day of the protest were used. Wouldn’t the two methods be more comparable if for the random sample, too, only tweets that were posted on the day of the protest would have been considered?"

We thank the reviewer on this final point and agree that this would constitute a better comparison. We therefore take a subsample of the original 10,000 user sample and include only those users who tweeted on the day of the Women’s March. We did not generate a new 10,000 user sample because doing so would mean having to re-estimate ideology scores at a later time point (which could bias findings as we are estimating ideology at a time less proximate to the actual protest, during which time the users may have followed different accounts).

REVIEWER 2

"The use of the term “revolutions” is a bit confusing in places, as it is used differently across fields. E.g., in line 37, I suspect the authors are seeking a point that is closer to questions of tipping points toward mass violence or direct action, rather than a full-scale overthrow of the existing order, which is implied by the term “revolution.”"

We thank the reviewer for drawing our attention to this. We now refer to mass mobilization events, explain what we mean by counter-revolutionaries, and refer to protest cascades that precipitate major uprisings.

"Leaving aside the difficulty of geocoding tweets more broadly, the authors should note whether they sought to categorize those who were incidentally in the location of protests. Large-scale protests are often in crowded urban environments, and those tweeting may be affected by the protests while not willfully participating in them. Naturally, whether those who are so caught up in protests should be included in a sample is a question of the research design and the research question at hand. But the assumption that the samples within a km of key central positions of the protests seems like it would catch a lot, most of the routes running through the densest business districts in each city, by design."

This was a useful piece of feedback, and this weakness obviously stood out to both reviewers. As for whether individuals might be “caught up” in the sample even if not in attendance at the protest, we now discuss this point in the main text, noting that whether or not they should be included will be a question of the research design and question, as this reviewer helpfully points out. It is also worth noting, however, that here it is less likely that individuals will be accidentally caught up in the sample because they are tweeting with the #WomensMarch identifying hashtag to make it into the sample in the first place. Of course, a casual observer or individual within the locale may also adopt the hashtag that they see being used on Twitter, even if not in attendance properly speaking at the march. While this may be the case, we find little evidence for this in our sample. As we detail in the Appendix section “Further sample characteristics,” we coded a random subsample of geolocated DC tweets and found little evidence that observers, opponents, or individuals otherwise incidentally at the scene are sizeable enough a minority to affect the findings. Our photo-coded sample provides an even harder test of this, where we code only those individuals as actual protestors if they satisfy a set of stringent criteria. That our geolocated and photo-coded samples are broadly comparable therefore adds confidence in our findings.

"Footnote 9. While it is reasonable to assume removal by Twitter was largely of bot accounts, it also seems likely that non-bot accounts were removed, and likely such removals weighed heavily toward the conservative side. By your own reasoning, we might likewise assume that tweet or account deletions were more likely to be by conservatives wishing to disassociate themselves with a “losing” political movement at some point after the 2019 elections."

This is an astute observation, and one that, as this reviewer notes, is consistent with our observation that individuals may not wish to associate themselves with losing or minority movements. We have added now to this footnote, making this point. Here, we note that deletion by members of the minority movement may have skewed findings but make clear that this principally applies to the hashtag sample, thereby providing additional weight to our warnings against the use of hashtag samples alone.

"It’s a bit easy to lose track of the process here. The figure in the appendix helps a bit here, but I should be able to easily discern this flow in the text as well. Am I correct in understanding that the hashtag sample was your starting point and the geolocated and photo samples were filtered from the hashtag (“random”) sample? How and when was the random sample obtained? Given the flat number, how was it sampled?"

We thank the reviewer for flagging this. We have now included more information in the main text on how the process of filtering the data. The starting dataset for each of the samples was a dataset of tweets including the hashtag #WomensMarch or similar. These were curated by Ruest (2017) and Littman & Park (2017). We also now include a revised workflow diagram that makes clearer the process of filtering these datasets to arrive at our final samples.

"The report of missing ideological scores (line 215) feels out of place given the lack of representation in the text of the n of the sample size. The percentages in Table 1 make this clearer, but the n for each of the samples should be highlighted early on (and likely listed in Table 1 as well, as total n for each sample)."

Similar to the above, we address this now in the main text. Beginning with the section entitled “Obtaining a sample of protestors,” through the subsequent three subsections we detail precisely how we filtered the original Twitter data to arrive at the final sample sizes for each. As noted above, the revised workflow diagram should ensure that this process is clear to the reader, as will the added Appendix sections describing the sources of missingness in the data (see Appendix section “Missingness”).

"Relatedly, am I to assume you assigned an ideological score with any number of follows of the ideologically inflected accounts more than zero? So that if someone, for example, follows Barack Obama, but no other listed account, they are coded (presumably as liberal)? I can follow this up in the cite, but it feels dangerous."

This is essentially correct. Some users may follow just one account, meaning that their ideology score will be measured with sizeable error. However, one qualification is also in order. When a user follows somebody like Barack Obama, who may be followed just by dint of popularity and political relevance for this interested in politics, the estimation procedure accounts for this through the inclusion of “elite random effects;” i.e., by incorporating a parameter that accounts for the popularity of a given account. To provide more information on the number of users who follow a small number of elites, we plot in the Appendix section “Following elite accounts” histograms for the number of elites our users follow in our hashtag (“Random”) and Photo-coded and Geolocated samples. The average (mean) number is fairly high for both, though is larger for the Geolocated/Photo-coded sample: 47 versus 28 (31 and 12 for median). Finally, we re-estimate our ideology distributions, excluding those users who follow fewer than five elite accounts. We plot the results in the Appendix, and see that the substantive conclusions remain identical. 

"The use of in-protest survey data as a comparator is perfectly reasonable, but calling these “ground truth” (ln 276) is problematic for many of the reasons you have already noted earlier in the manuscript. Protest survey data come with their own significant biases. One might make some guesses as to systematic error here that run pretty close to things like demographic data. Even in a relatively safe setting, women may be less likely to answer unsolicited questions by someone approaching them in public, for example. But more broadly, these are multiple attempts to ascertain a ground-truth that each approach is attempting to approximate. Comparing the multiple attempts against one another is natural, but assigning the survey data as the gold standard would require you to more clearly indicate why you are assuming this to be the case."

We think this is a fair criticism and have now removed mention of “ground-truth,” and refer instead to the in-protest surveys as our “benchmark.” We nonetheless view these as a valid benchmark and as high quality data, noting the low refusal rates and close correspondence between these two independent sampling efforts.

"I feel like this drops things off rapidly at the end. This may be a matter of the brevity of the report. However, the question of (for example) differences in age could be explained in multiple ways: most especially, differences in age distribution among twitter users (or SM users mid-protest, more specifically) or a classification model that introduces systematic errors in approximating age. The latter could be ascertained by surveying twitter users for their age, either online (though tracking twitter users to survey them can be tricky and invasive) or by surveying protestors in person to ascertain whether they are tweeting (and potentially linking samples directly). In any case, it would be very helpful for the discussion to open up possibilities of extending the work undertaken thus far."

Thank you for this. We agree that the discussion was too brief and we did not do enough to reflect on the findings. We have now made changes to the first several paragraphs of the Discussion, detailing more fully the potential explanations for the discrepancy between our samples. We also add a section about future research, and incorporate mention of the reviewer’s valuable suggestion to include requests for social media handles in future in-protest surveys.

---

## [Decision Letter · Decision Letter 1]

2 Nov 2021

Faces in the Crowd: Twitter as Alternative to Protest Surveys

PONE-D-21-07964R1

Dear Dr. Frey,

We’re pleased to inform you that your manuscript has been judged scientifically suitable for publication and will be formally accepted for publication once it meets all outstanding technical requirements.

Kind regards,

Barbara Guidi

Academic Editor

PLOS ONE

Additional Editor Comments (optional):

Reviewers' comments:

Reviewer's Responses to Questions

**Comments to the Author**

1. If the authors have adequately addressed your comments raised in a previous round of review and you feel that this manuscript is now acceptable for publication, you may indicate that here to bypass the “Comments to the Author” section, enter your conflict of interest statement in the “Confidential to Editor” section, and submit your "Accept" recommendation.

Reviewer #1: All comments have been addressed

2. Is the manuscript technically sound, and do the data support the conclusions?

Reviewer #1: Yes

3. Has the statistical analysis been performed appropriately and rigorously? 

Reviewer #1: Yes

4. Have the authors made all data underlying the findings in their manuscript fully available?

Reviewer #1: No

5. Is the manuscript presented in an intelligible fashion and written in standard English?

Reviewer #1: Yes

6. Review Comments to the Author

Reviewer #1: Dear Authors,

Thank you for submitting your revised version of the manuscript. All comments have been addressed adequately and in much detail. The manuscript has improved greatly and I therefore regard it as acceptable for publication.

Kind regards

the Reviewer

7. PLOS authors have the option to publish the peer review history of their article (what does this mean?). If published, this will include your full peer review and any attached files.

Reviewer #1: No

---

## [Editor Report · Acceptance letter]

8 Nov 2021

PONE-D-21-07964R1 

Faces in the crowd: Twitter as alternative to protest surveys 

Dear Dr. Frey:

I'm pleased to inform you that your manuscript has been deemed suitable for publication in PLOS ONE. Congratulations! Your manuscript is now with our production department. 

Kind regards, 

on behalf of

Dr. Barbara Guidi 

Academic Editor

PLOS ONE